# The Endometriotic Neoplasm Algorithm for Risk Assessment (e-NARA) Index Sheds Light on the Discrimination of Endometriosis-Associated Ovarian Cancer from Ovarian Endometrioma

**DOI:** 10.3390/biomedicines10112683

**Published:** 2022-10-24

**Authors:** Naoki Kawahara, Ryuji Kawaguchi, Tomoka Maehana, Shoichiro Yamanaka, Yuki Yamada, Hiroshi Kobayashi, Fuminori Kimura

**Affiliations:** Department of Obstetrics and Gynecology, Nara Medical University, Kashihara 634-8522, Japan

**Keywords:** ovarian endometrioma, endometriosis-associated ovarian cancer, magnetic resonance imaging, MR relaxometry, the R2 value, the endometriotic neoplasm algorithm for risk assessment (e-NARA) index

## Abstract

Background: Magnetic resonance (MR) relaxometry provides a noninvasive tool to discriminate endometriosis-associated ovarian cancer (EAOC) from ovarian endometrioma (OE) with high accuracy. However, this method has a limitation in discriminating malignancy in clinical use because the R2 value depends on the device manufacturer and repeated imaging is unrealistic. The current study aimed to reassess the diagnostic accuracy of MR relaxometry and investigate a more powerful tool to distinguish EAOC from OE. Methods: This retrospective study was conducted at our institution from December, 2012, to May, 2022. A total of 150 patients were included in this study. Patients with benign ovarian tumors (*n* = 108) mainly received laparoscopic surgery, and cases with suspected malignancy (*n* = 42) underwent laparotomy. Information from a chart review of the patients’ medical records was collected. Results: A multiple regression analysis revealed that the age, the tumor diameter, and the R2 value were independent malignant predicting factors. The endometriotic neoplasm algorithm for risk assessment (e-NARA) index provided high accuracy (sensitivity, 85.7%; specificity, 87.0%) to discriminate EAOC from OE. Conclusions: The e-NARA index is a reliable tool to assess the probability of malignant transformation of endometrioma.

## 1. Introduction

Ovarian cancer is the fifth leading cause of cancer-related death in women [1]. This disease cannot be diagnosed in the early stages and is called the silent killer [2,3,4]. As such, most ovarian cancer cases are diagnosed at advanced stages [5,6,7], and over 185,000 deaths due to this disease are reported annually worldwide [8,9]. Ovarian cancer is divided into epithelial, germ cell, and sex cord-stromal tumors, and, of these, epithelial ovarian cancer has the highest rate [10,11]. Epithelial ovarian cancer can be divided into two categories, designated as types 1 and 2 [12,13,14], by molecular genetics and morphologic characteristics. Type 1 tumors show a stepwise progression (adenoma–carcinoma sequence), which comprises endometriosis-associated ovarian cancer (EAOC), such as clear cell carcinoma (CCC) and low-grade endometrioid carcinoma, as well as mucinous carcinoma and low-grade serous carcinoma [15,16]. Type 2 tumors range from the normal epithelium to precursor lesions, and, finally, to high-grade serous and endometrioid carcinoma, malignant mixed mesodermal tumors (carcinosarcomas), and undifferentiated carcinoma [15,17]. The former shows slow, and the latter fast, progression to advanced stages [18,19].

Ovarian endometriosis is defined as the presence of endometrial glands and stroma outside of the uterus, and it is most often detected in the pelvic peritoneum and ovaries [20,21,22]. Repeated hemorrhages in the peritoneum or ovaries may contribute to several symptoms of dysmenorrhea [23,24,25], chronic pelvic pain [26,27,28,29], and infertility [30,31,32,33], which negatively affect the patient’s quality of life [34,35,36]. Epidemiologically, endometriosis has been reported to increase the risk of EAOC, such as EC, CCC, low-grade serous carcinoma, and seromucinous neoplasms [37,38,39]. CCC and EC of the ovary are the two most common types of ovarian cancer, which arise from endometriosis [38,39].

In general, the presence of mural nodules and papillary projections is considered to constitute evidence of malignancy [40]. These can be seen in either OE or EAOC, which can pose a challenging diagnostic dilemma to clinicians [41,42]. Therefore, we investigated how to discriminate EAOC from OE, and showed that total iron levels of cyst fluid can discriminate EAOC from ovarian endometrioma (OE), with a cut-off point of 64.8 mg/L (sensitivity, 85%; specificity, 98%) [43]. Magnetic resonance (MR) relaxometry, which can noninvasively measure cyst fluid iron concentration, can discriminate with a cut-off point of 12.1 (sensitivity, 86%; specificity, 94%) [44]. Moreover, we showed a novel predictive tool in the R2 predictive index, which requires tumor diameter and serum CEA level. This model had good efficacy to detect the malignant transformation of endometrioma (i.e., EAOC) without MRI, with good accuracy (sensitivity, 82%; specificity, 68%), and is useful in following up outpatients [45,46]. MR relaxometry has exhibited a limitation in discriminating malignancy for preoperative assessment [i.e., false positive (FP) or false negative (FN)].

The current study aimed to reassess the diagnostic accuracy of MR relaxometry and investigate both a more powerful and non-invasive tool to discriminate EAOC from OE.

## 2. Materials and Methods

### 2.1. Patients

A list of patients with primary, previously untreated, histologically-confirmed ovarian tumors, who were treated at Nara Medical University Hospital between December, 2012, and May, 2022, was generated from our institutional registry. We retrospectively included in this study the following cases of OE as benign ovarian tumors and EAOC cases as malignant tumors. Patients who were over 20 years old at the time of surgery and who consented to, and received, magnetic resonance imaging (MR imaging) after hospitalization were included in the current cohort. Patients who were under 20 years old, contraindicated for MR imaging, prone to claustrophobia, or who refused to undergo MR imaging after hospitalization were excluded. All of the OE and EAOC cases were histologically confirmed. Written consent for the use of the patients’ clinical data for research was obtained at the first hospitalization, and, after approval by the Ethics Review Committee of the Nara Medical Hospital, the opt-out form was provided through our institutional homepage. A total of 150 patients were included in the current cohort. One hundred and eight patients were benign OE cases and forty-two patients were malignant cases. No patients had undergone chemotherapy or radiotherapy for the ovarian tumors prior to treatment. Patients with OE mainly received laparoscopic surgery, and the patients suspected of harboring malignant tumors underwent laparotomy. The following factors were collected through a chart review of the patients’ medical records: age, body mass index (BMI), parity, postoperative diagnosis, including FIGO (The International Federation of Gynecology and Obstetrics) stage, tumor diameter, menopausal status, and pre-treatment blood test results, including carbohydrate antigen125 (CA125), carbohydrate antigen 19-9 (CA 19-9), and carcinoembryonic antigen (CEA) as a tumor marker.

### 2.2. Tumor Imaging and Diagnoses

All patients first visited the outpatient clinic and underwent internal examination, including ultrasound, followed by routine MR imaging using T1W and T2W sequences. Tumor diameter was recorded as the largest diameter among axial, sagittal, and coronal imaging. Patients were largely diagnosed with OE or EAOC by MRI, and this was confirmed by histological examination, using surgically removed tissue, by at least two pathologists who were blinded to the study. The R2 values were obtained by a 3T system (Magnetom Verio or Skyla, Siemens Healthcare, Erlangen, Germany). After the routine clinical MR imaging, the registered patients underwent MR relaxometry using the single-voxel acquisition mode sequence at multiple echo times and by fitting an exponential decay to the echo amplitude at different multiple echo times [47]. A parameter R2 value (s-1) was calculated using a high-speed T2 *-corrected multi-echo MR sequence (HISTO) by the 3T–MR system in vivo and ex vivo, which has been previously described [48,49]. The HISTO sequence was based on the single voxel steam sequences that could be used for relative fat quantification in the liver [50]. This sequence allows estimation of liver iron deposition, since the T2 of water changes with iron concentration. The pulse sequence design and programming were done with an imaging platform (Siemens Medical Systems, Erlangen, Germany) and applied to the 3T system. The sequence had a fixed number of five measurements with different TEs, which were as follows: 12, 24, 36, 48, and 72 ms. The typical protocol was performed in breath-hold, with a total acquisition time of 15 sec. The repetition time (TR) was fixed to 3000 ms, which proved to be enough to compensate for the effects of signal saturation, while maintaining an acceptable acquisition time. A 15 × 15 × 15-mm spectroscopy voxel (VOI) was placed to select a region encompassing the liquid portion, but not the solid portion, of the cyst lumen. The fluid from the largest cyst was measured if there were any patients who had more than one cyst. The VOI was located in the center of the OE or EAOC cyst by a radiologist who specializes in female pelvic MR imaging.

### 2.3. Statistical Analysis

Analyses were performed using SPSS version 25.0 (IBM SPSS, Armonk, NY, USA). The differences of each factor, including the CPH index, the ROMA index, and the R2 predictive index, among groups were compared using a Mann–Whitney U test or Kruskal–Wallis one-way ANOVA test. A receiver operating characteristic (ROC) curve analysis was performed to determine the cut-off value for predicting malignant ovarian tumors. The cut-off value was based on the highest Youden index (i.e., sensitivity + specificity − 1). We next used a logistic regression analysis to assess the risk factors for malignant ovarian tumors (i.e., EAOC). A two-sided *p* < 0.05 was considered as indicating a statistically significant difference.

## 3. Results

### 3.1. Patients

From December, 2012, to May, 2022, a total of 150 patients were included in this study. The benign and malignant cases were 108 and 42 in number, respectively. The demographic and clinical characteristics of the combined cohort are outlined in Table 1.

Cases diagnosed as borderline or harboring atypical cells in the cystic epithelial tissue were included in malignant cases. In this cohort, there was significant differentiation in age, maximum tumor diameter, and menopausal status. Table 2 shows the distribution of peripheral blood cells. The platelet counts, lymphocyte (% and counts), and monocyte (%) reached significant differentiation between a benign tumor and a malignant tumor.

Table 3 shows the distribution of serum markers and blood coagulation examination in peripheral blood cells and the R2 value obtained by MR relaxometry. The carcinoembryonic antigen (CEA), C-reactive protein (CRP), albumin, D-dimer, activated partial thromboplastin time (APTT), and R2 showed significant differentiation between benign and malignant tumors.

### 3.2. The Efficacy of Each Factor in Discriminating between OE and EAOC

The results of the ROC curve analysis, based on the detection of malignant tumors, are shown in Table 4. The optimal cut-off value was determined by analyzing the ROC curve among malignant ovarian tumors and OEs. The ROC analysis showed the same result as peripheral blood cell distribution, serum markers, and blood coagulation examination results (Table 2 and Table 3). The R2 value showed the best sensitivity, and age and cyst size showed the top two specificities (Table 4, Figure 1).

### 3.3. The Independent Factors in Discriminating OE and EAOC

A multivariate analysis confirmed that age, cyst size, and the R2 value were extracted as independent factors for predicting malignant tumors (hazard ratio (HR): 14.35, 95% confidence interval (CI): 2.89–71.04, *p* < 0.001; HR: 14.40, 95% CI: 3.26–63.51, *p* < 0.001; HR: 10.23, 95% CI: 2.60–40.20, *p* = 0.001, respectively) (Table 5).

### 3.4. The Efficacy of Endometriotic Neoplasm Algorithm for Risk Assessment (e-NARA) Index in Discriminating OE and EAOC

We created the endometriotic neoplasm algorithm for risk assessment (e-NARA) index, which was calculated using the following Equation (1):e-NARA index = −3.836 + 2.664 × [age(year)/10] + 2.667 × LN [Tumor diameter(mm)] + 2.326 × [10/R2](1)

LN = natural log function.

We next assessed the efficacy of the e-NARA Index in discriminating between OE and EAOC. The result of the ROC curve analysis, based on discriminating EAOC from OE, is shown in Figure 2A. The cut-off value from the above formula was 21.36 (sensitivity, 85.7%; specificity, 87.0%; AUC = 0.928, *p* < 0.001) (Figure 2A,B). When setting the cut-off value as 17.97, sensitivity and specificity were 100.0% and 48.1%, respectively, and when the cut-off value was set as 25.91, sensitivity and specificity were 38.1% and 100.0%, respectively (Figure 2B).

### 3.5. The Sub-Group Analysis of Malignant Tumor Showed the E-NARA Index Increased Stepwise

When the malignant tumors were divided into those with atypia/borderline tumor and those with advanced malignant tumor, age, tumor diameter, and R2 value changed stepwise. Notably, age and R2 could distinguish atypia/borderline tumor from benign OE and from atypia/borderline tumor with significant differentiation (*p* = 0.029 and *p* = 0.046, respectively) (Figure 3A,C). The e-NARA index, which consisted of the above factors, showed significant differentiation in discriminating atypia/borderline tumor from benign OE, and advanced malignant tumor from benign OE, with optimal cut-off values of 19.89 and 21.36, respectively (Table 6, Figure 3D). When comparing benign OE and atypia/borderline tumor, under 19.89 corresponded to benign tumor (Figure 3(D-a)). The cut-off value discriminating advanced malignant tumor from benign OE was the same value, as shown in Figure 2(B-a) and Figure 3(D-b).

## 4. Discussion

We previously reported that MR relaxometry could be a noninvasive preoperative prediction tool and showed a favorable predictive accuracy for malignant transformations, with sensitivity and specificity of 86% and 94%, respectively [44]. In the current study, MR relaxometry showed a similar sensitivity to (85.7%), but lower specificity (80.6%) than, the previous reports. This result could have been influenced by the accumulative effect of the cases. MR relaxometry has diagnostic limitations in clinical use in the case of FP or FN. The e-NARA index improved the specificity (87.0%).

Ovarian tumors are diagnosed mainly as benign or malignant by transvaginal ultrasound because of its low cost and easily operable characteristics. However, this device has yielded to enhanced MR imaging, because of its poor subjectivity. To improve its weak point, the International Ovarian Tumor Analysis (IOTA) Group developed a system of standardization in the characterization of adnexal masses [51]. Lee Cohen Ben-Meir et al. investigated this method in OE, or its associated malignant tumor, EAOC, and reported that this method could discriminate malignant tumors with high sensitivity [52]. Since our study showed relatively high specificity, the IOTA system is recommended to evaluate ovarian tumors for screening, and the e-NARA to validate.

Among previously reported indexes, the risk of ovarian malignancy algorithm (ROMA) index and the Copenhagen (CPH) index were the two major predictive tools that use serum markers and age, or menopausal status, in discriminating malignant ovarian tumor from benign [53,54]. Our reports also showed tumor diameter and serum CEA level could discriminate EAOC from OE without MRI [45,46]. Similar to these indexes, the e-NARA index included one of these factors, such as age, and tumor diameter improved the diagnostic accuracy. In discriminating malignant from benign tumors, relying on only one indicator (i.e., MR relaxometry) could be inadequate, and indexes using multilateral indicators, as above, should be required.

CEA is reported as an independent predictor for identifying epithelial ovarian cancer and ovarian metastases [55]. Further studies found that the cut-off value of CEA in the differential diagnosis of primary ovarian tumors and metastatic ovarian cancer was 2.33 μg/L [56]. In this cohort, CEA showed good diagnostic efficacy in univariate analysis. However, it did not achieve significant differentiation in multivariate analysis. The serum CEA level could exert its ability in predicting the R2 value, rather than in discriminating EAOC from OE with the real R2 value.

In recent years, inflammatory reactions in the tumor microenvironment have been shown to play an important role in tumor development and progression [57,58]. Peripheral leukocytes, neutrophils, lymphocytes, platelets, and acute-phase proteins contribute to the inflammatory response and can be detected easily. A number of studies have demonstrated that inflammatory response factors are related to the survival of patients with cancer who have been surgically treated [59,60,61,62,63]. In the current study, inflammatory factors, such as elevated monocytes, platelets, and CRP, and decreased lymphocytes and albumin, showed good diagnostic efficacy at univariate analysis, which was comparable to previous reports [64,65,66,67,68,69]. Similar to the above malignant tumors, OE also induces severe inflammatory responses [70,71,72]. The inflammatory response between OE and EAOC should be different.

Finally, the current study supports a scenario of the 2-step malignant transformation model which Hiroshi Kobayashi et al. hypothesized [73]. In the first step, excess hemoglobin and iron species, produced by autoxidation and the Fenton reaction cause oxidative damage, which results in DNA damage and mutations. In the second step, reduced iron content and increased antioxidant protection could help in cell survival and the tumorigenic effect of endometriotic cells. In the current study, the sub-group analysis showed that the R2 value, which reflects the iron concentration of the cyst fluid, reduced stepwise in the order of benign OE to malignant tumor. The iron species could play a key role in causing malignant transformation of OE, and further investigation into the balance of iron species and antioxidant protection is required.

This study had some limitations. The first limitation was possible selection bias, due to the nature of retrospective study. To overcome this bias, a multi-center prospective cohort study is now proceeding (UMIN000034969). Second, the newly reported tumor marker of tissue factor pathway inhibitor 2 (TFPI2) was not assessed [74,75]. Finally, we did not compare diagnostic efficacy among the IOTA classification, ROMA index, the CPH index, and the e-NARA index. To assess the efficacy of the above tumor marker and these indexes, a prospective study is needed.

## 5. Conclusions

In conclusion, the e-NARA index improved diagnostic efficacy in discriminating EAOC from OE and could provide clinicians with reliable evidence to diagnose EAOC.

## Figures and Tables

**Figure 1 biomedicines-10-02683-f001:**
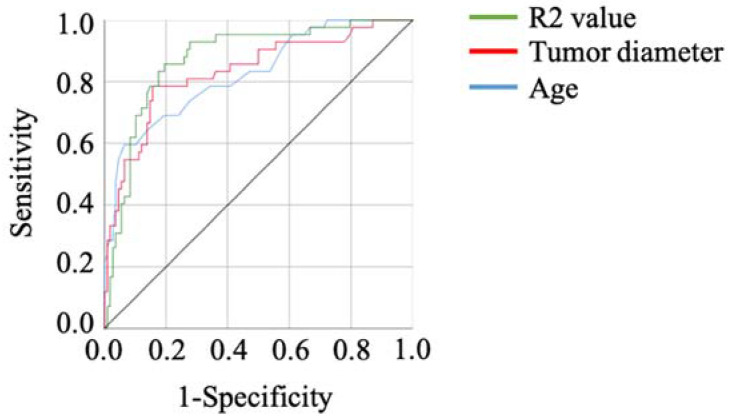
The ROC curves of the factors showing the top three specificities in the current cohort. The R2 value showed a high AUC.

**Figure 2 biomedicines-10-02683-f002:**
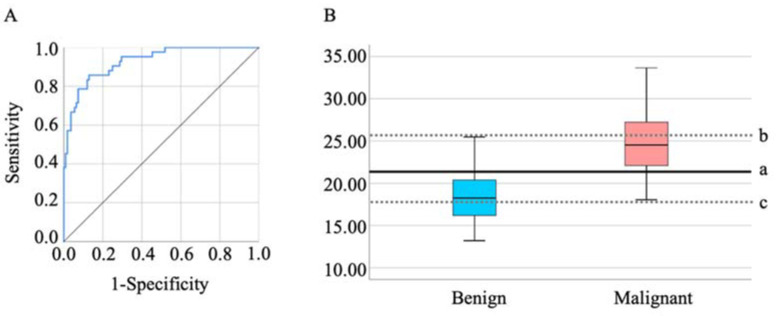
(**A**) The ROC curves of the e-NARA index. The optimal cut-off value was 21.36 (**B**-**a**). To make each specificity and sensitivity 100%, the cut-off values of the e-NARA index were 25.91 (**B**-**b**) and 17.97 (**B**-**c**), respectively.

**Figure 3 biomedicines-10-02683-f003:**
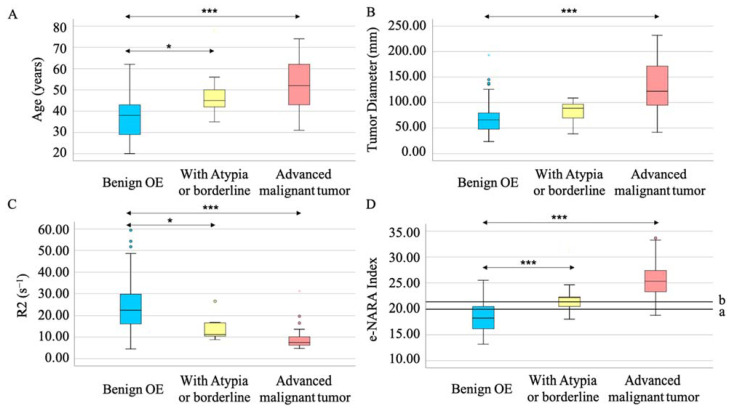
The e-NARA index, which consisted of age (**A**), tumor diameter (**B**), and R2 value (**C**) could discriminate between atypia/borderline tumor from OE (**D**-**a**) and advanced malignant tumor from benign OE (**D**-**b**). Circles and stars represent outlier (**B**,**C**). *** *p* < 0.001 and * *p* < 0.05 vs. benign OE.

**Table 1 biomedicines-10-02683-t001:** Demographic and clinical characteristics of the current cohort.

	Benign Tumor	Malignant Tumor	*p*-Value
Number	*n* = 108	*n* = 42	
Age (years)			
Median (range)	38.00 (20–62)	50.00 (31–78)	
Mean ± SD	37.04 ± 8.22	51.14 ± 12.39	<0.001
BMI			
Median (range)	21.43 (14.88–38.03)	21.78 (17.01–39.48)	
Mean ± SD	21.66 ± 3.69	23.10 ± 5.08	0.238
Parity			
0	58	20	
≥1	50	22	0.586
FIGO stage	-	I (*n* = 35), II (*n* = 6), III (*n* = 1)	
Subtype	Endometriosis (*n* = 108)	CCC (*n* = 18)	
		Endometrioid carcinoma (*n* = 13)	
		Seromucinous (*n* = 8)	
		With atypical cells (*n* = 2)	
		CCC + Endomtrioid (*n* = 1)	
Cyst size (mm)			
Median (range)	66.14 (23.53–193.00)	110.00 (38.99–231.92)	
Mean ± SD	66.72 ± 28.06	121.05 ± 50.76	<0.001
Menopause			
Yes	4	18	
No	104	24	<0.001

BMI body mass index, FIGO The International Federation of Gynecology and Obstetrics, CCC clear cell carcinoma.

**Table 2 biomedicines-10-02683-t002:** Distributions of peripheral blood cells and serum inflammatory values in the current cohort.

	Benign Tumor	Malignant Tumor	*p*-Value
Number	*n* = 108	*n* = 42	
Hb (g/mL)			
Median (range)	12.70 (7.50–14.80)	12.55 (7.20–15.20)	
Mean ± SD	12.47 ± 1.27	12.26 ± 1.68	0.652
Platelet (×10^4^/µL)			
Median (range)	25.90 (10.10–42.20)	28.70 (16.20–57.80)	
Mean ± SD	26.36 ± 5.80	30.01 ± 8.74	0.011
WBC (×10^2^/µL)			
Median (range)	61.00 (27.00–182.00)	70.00 (28.00–149.00)	
Mean ± SD	64.84 ± 22.71	69.04± 25.00	0.267
Neutrophils (%)			
Median (range)	61.80 (41.00–91.50)	67.70 (41.10–94.10)	
Mean ± SD	63.08 ± 10.01	66.07 ± 12.19	0.134
Neutrophils (×10^2^/µL)			
Median (range)	37.57 (13.12–156.46)	44.41 (12.29–122.33)	
Mean ± SD	42.90 ± 23.18	45.09 ± 23.43	0.517
Lymphocytes (%)			
Median (range)	29.20 (7.00–43.00)	21.80 (2.70–45.60)	
Mean ± SD	27.44 ± 8.59	23.95 ± 10.05	0.035
Lymphocytes (×10^2^/µL)			
Median (range)	16.41 (6.30–28.22)	13.67 (3.51–29.18)	
Mean ± SD	16.62± 4.59	13.88 ± 4.48	0.002
Monocytes (%)			
Median (range)	5.90 (1.40–12.20)	6.50 (1.40–11.70)	
Mean ± SD	6.10 ± 1.65	6.96 ± 2.15	0.016
monocytes (×10^2^/µL)			
Median (range)	3.66 (1.34–9.28)	3.96 (1.82–8.10)	
Mean ± SD	3.87± 1.39	4.24 ± 1.42	0.200

Hb hemoglobin, WBC white blood cells.

**Table 3 biomedicines-10-02683-t003:** Serum markers and blood coagulation examination in peripheral blood cells and R2 value obtained by MR relaxometry.

	Benign Tumor	Malignant Tumor	*p*-Value
Number	*n* = 108	*n* = 42	
CEA (ng/mL)			
Median (range)	0.90 (0.30–4.20)	1.40 (0.40–67.6)	
Mean ± SD	1.12 ± 0.81	3.68 ± 10.39	<0.001
CA125 (U/mL)			
Median (range)	63.50 (9.00–15.04 × 10^2^)	46.00 (8.00–10.59 × 10^3^)	
Mean ± SD	103.90 ± 164.71	640.92 ± 1764.63	0.496
CA 19-9 (U/mL)			
Median (range)	24.00 (1.00–4.74 × 10^2^)	26.00 (1.00–19.94 × 10^4^)	
Mean ± SD	40.47 ± 61.32	5229.70 ± 31,108.10	0.164
CRP (mg/dL)			
Median (range)	0.02 (0.00–12.00)	0.10 (0.00–13.40)	
Mean ± SD	0.35 ± 1.32	0.88 ± 2.43	0.008
Albumin (g/dL)			
Median (range)	4.50 (3.60–5.20)	4.40 (3.50–5.20)	
Mean ± SD	4.47 ± 0.25	4.34 ± 0.31	0.025
D-dimer (µg/mL)			
Median (range)	0.60 (0.40–4.50)	0.80 (0.40–17.30)	
Mean ± SD	0.77± 0.54	2.39 ± 3.77	0.009
APTT (second)			
Median (range)	28.30 (23.00–48.30)	27.30 (24.20–36.30)	
Mean ± SD	28.79 ± 3.44	27.59 ± 2.68	0.023
R2 (s^−1^)			
Median (range)	22.30 (4.53–59.42)	8.38 (4.80–31.22)	
Mean ± SD	23.68 ± 11.19	10.12 ± 5.58	<0.001

CEA carcinoembryonic antigen, CA125 carbohydrate antigen125, CA 19-9 carbohydrate antigen 19-9, CRP C-reactive protein, APTT activated partial thromboplastin time.

**Table 4 biomedicines-10-02683-t004:** The cut-off values discriminating EAOC from benign OE in the current cohort.

	AUC	*p*-Value	Cut-Off Value	Sensitivity	Specificity	PPV	NPV
Age (years)	0.825	<0.001	47.50	0.595	0.935	78.12	85.59
Cyst size (mm)	0.833	<0.001	85.74	0.786	0.843	66.00	91.00
CEA (ng/mL)	0.711	<0.001	0.950	0.756	0.583	50.82	80.76
Lymphocytes (%)	0.622	0.035	22.50	0.571	0.759	48.78	81.48
Lymphocytes (×10^2^/µL)	0.680	0.002	16.06	0.800	0.540	40.58	86.79
Monocytes (%)	0.640	0.016	7.65	0.371	0.862	52.00	77.32
Platelet (×10^4^/µL)	0.635	0.011	29.25	0.500	0.743	43.75	78.78
CRP (mg/dL)	0.638	0.009	0.14	0.476	0.757	44.44	78.00
Albumin (g/dL)	0.619	0.026	4.25	0.381	0.825	48.48	75.47
D-dimer (µg/mL)	0.648	0.010	0.75	0.556	0.735	47.61	79.22
APTT (second)	0.622	0.023	27.75	0.619	0.611	41.27	78.37
R2 (s^−1^)	0.875	<0.001	13.76	0.857	0.806	63.15	93.54

CEA carcinoembryonic antigen, CRP C-reactive protein, APTT activated partial thromboplastin time, PPV positive predictive value, NPV negative predictive value, AUC area under curve. The R2 value was calculated as a fraction.

**Table 5 biomedicines-10-02683-t005:** Univariate and Multivariate analysis of the predictive factors of EAOC in the current cohort.

		Univariate Analysis		Multivariate Analysis	
		Risk Ratio (95% CI)	*p*-Value	Risk Ratio (95% CI)	*p*-Value
Age	≤47.50	1.00 (referent)		1.00 (referent)	
(years)	>47.50	21.21 (7.93–56.71)	<0.001	14.35 (2.89–71.04)	0.001
Cyst size	≤85.74	1.00 (referent)		1.00 (referent)	
(mm)	>85.74	19.62 (7.97–48.31)	<0.001	14.40 (3.26–63.51)	<0.001
CEA	≤0.95	1.00 (referent)			
(ng/mL)	>0.95	4.34 (1.84–10.18)	0.001		
Lymphocytes	>22.50	1.00 (referent)			
(%)	≤22.50	4.19 (1.82–9.61)	0.001		
Lymphocytes	>16.05	1.00 (referent)			
(×10^2^/µL)	≤16.05	4.48 (1.77–11.36)	0.002		
Monocytes	≤7.65	1.00 (referent)			
(%)	>7.65	3.69 (1.47–9.24)	0.005		
Platelet	≤29.25	1.00 (referent)			
(×10^4^/µL)	>29.25	2.88 (1.36–6.09)	0.005		
CRP	≤0.14	1.00 (referent)			
(mg/dL)	>0.14	2.83 (1.33–6.03)	0.007		
Albumin	>4.25	1.00 (referent)			
(g/dL)	≤4.25	2.89 (1.28–6.53)	0.010		
D-dimer	≤0.75	1.00 (referent)			
(µg/mL)	>0.75	3.46 (1.52–7.85)	0.003		
APTT	>27.75	1.00 (referent)			
(second)	≤27.75	2.54 (1.20–5.37)	0.014		
R2	>13.76	1.00 (referent)		1.00 (referent)	
(s^−1^)	≤13.76	24.85 (9.26–66.69)	<0.001	10.23 (2.60–40.20)	0.001

CEA carcinoembryonic antigen, CRP C-reactive protein, APTT activated partial thromboplastin time. The R2 value was calculated as a fraction.

**Table 6 biomedicines-10-02683-t006:** The cut-off values among benign OE, atypia or borderline tumor, and advanced malignant tumor in the current cohort.

	AUC	*p*-Value	Cut-Off Value	Sensitivity	Specificity	PPV	NPV
Benign OE vs. Atypia or borderline tumor	0.846	0.001	19.89	0.889	0.713	20.51	98.71
Benign OE vs. Advanced malignant tumor	0.951	<0.001	21.36	0.909	0.870	68.18	96.90

OE ovarian endometrioma.

## Data Availability

The data presented in this study are available on request from the corresponding author.

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
