# Peer review of "The Endometriotic Neoplasm Algorithm for Risk Assessment (e-NARA) Index Sheds Light on the Discrimination of Endometriosis-Associated Ovarian Cancer from Ovarian Endometrioma"

_biomedicines, 2022, doi:10.3390/biomedicines10112683_

Round 1
Reviewer 1 Report
Discrimination of Endometriosis-Associated Ovarian Cancer from Ovarian Endometrioma is not easy, it is a challenge, precisely because according to an accurate diagnosis the appropriate therapeutic protocol is established.
I have some questions:
1. Preoperative diagnostic evaluation of benignity versus malignancy was confirmed in all cases on resection specimens at HP examination?
2. Cancers were identified, confirmed HP on resection specimens, in cases presumed preoperatively as benign and in which laparoscopic surgery was performed? And reciprocally, all cases in which the preoperative diagnosis was of malignancy, was confirmed at the HP examination of the resection specimens?
Author Response
Thank you very much for your detailed and kind review, and for giving us the chance to revise our manuscript. We felt very appreciated by the reviewer’s comments, and I followed all of your suggestions. It was very suggestive and improved remarkably our manuscript.
Answer to the reviewer's question, 1. All resected specimens were evaluated and diagnosed in the department of pathology in our hospital. 2. In the current cohort, there was no case that was preoperatively diagnosed as benign and postoperatively diagnosed as cancer. But there have been some cases who have conducted surgery as benign in other hospitals and diagnosed as cancer and were referred to our hospital. All cases, in our hospital, preoperatively diagnosed as malignancy are conducted with a rapid intraoperative examination during surgery and in cases confirmed advanced cancer is conducted radical surgery according to ovarian cancer treatment.Reviewer 2 Report
This Japanese retrospective study evaluated the diagnostic utility of MR relaxometry and the reliability of e-NARA index in distinguishing benign from malignant ovarian neoplasms. The study is problematic for many reasons. MR is rarely used in routine practice as it is more expensive than US which represents a reliable tool in distinguishing benign and malignant lesions. The authors fail to mention IOTA criteria. Furthermore, borderline tumours should be evaluated as a separate group. The methods are not clearly described, exclusion criteria are not presented and the limitations of this study are not properly described.
Author Response
Thank you very much for your detailed and kind review, and for giving us the chance to revise our manuscript. We felt very appreciated by the reviewer’s comments, and I followed all of your suggestions. It was very suggestive and improved remarkably our manuscript.
I changed the inclusion and exclusion criteria and supplemented the limitation.
Of course, the IOTA classification has excellent efficacy to suspect malignancy at a low cost. But some cases which are too difficult cases to decide the operation method are exist. To improve the above, a suspicious or difficult case should be analyzed with another modality such as MR imaging or such method as the ROMA index, CPH index, and e-NARA index.
We evaluated as a separate group and added the results according to the reviewer. These results included very important and novel evidence, namely precancer cyst fluid showed a lower R2 value than OE, which indicates the iron concentration of precancerous tumor is lower than OE. As a rare characteristic of atypia or borderline tumor, this has been unknown. The hypothesis that iron species play an important role in tumorigenesis of OE was partly proved by this study.